

# EEG based assessment of stress in horses: a pilot study

Nora V. de Camp[1,2,*], Mechthild Ladwig-Wiegard[2,*],
Carola I.E. Geitner[2], Jürgen Bergeler[2,3] and Christa Thöne-Reineke[2]

[1] Behavioral Physiology, Institute of Biology, Humboldt-Universität zu Berlin, Berlin, Germany
[2] Institute of Animal Welfare, Animal Behavior and Laboratory Animal Science, Freie Universität Berlin, Berlin, Germany
[3] Biology, Humboldt-Universität zu Berlin, Berlin, Germany
* These authors contributed equally to this work.

## ABSTRACT

As has been hypothesized more than 20 years ago, data derived from Electroencephalography (EEG) measurements can be used to distinguish between behavioral states associated with animal welfare. In our current study we found a high degree of correlation between the modulation index of phase related amplitude changes in the EEG of horses ($n = 6$ measurements with three different horses, mare and gelding) and their facial expression, measured by the use of the horse grimace scale. Furthermore, the pattern of phase amplitude coupling (PAC) was significantly different between a rest condition and a stress condition in horses. This pilot study paves the way for a possible use of EEG derived PAC as an objective tool for the assessment of animal welfare. Beyond that, the method might be useful to assess welfare aspects in the clinical setting for human patients, as for example in the neonatal intensive care unit.

## INTRODUCTION

Is Electroencephalography (EEG) a useful tool to assess welfare in horses? Animal welfare and animal well-being is part of controversial discussions. This happens in different contexts, be it socio-political or ethical discourses, factory farming and food production, animals in private husbandry, animal-assisted therapy, zoos, wildlife or animal experiments. They all have in common that both forming and exchange of opinions are often based on emotions rather than on scientific findings. The assessment of welfare and well-being of animals is sometimes made by how humans feel when they find animals in certain situations. Besides the lack of ethological knowledge, in many areas related to animal husbandry or the use of animals, there are none or only imprecise legal regulations. In the field of laboratory animal science, legal matters are more closely regulated, both in the context of authorization and in connection with surveillance. The commencement of Directive 63/2010/EU enforced efforts concerning animal welfare on a national and European level and specified personal and institutional prerequisites. People working in this field have to prove their expertise and have to be continuously

Corresponding author
Nora V. de Camp,
ndecamp@jpberlin.de

educated. Housing facilities and experiments have to be approved and procedures have to be ethically justified. Serious and extensive efforts have to be made in order to replace, reduce and refine (3Rs) experimental procedures performed on the animals or their housing conditions. There must be a close documentation of all interventions and a scientifically reasoned assessment and classification in degrees of severity according to the impairment of the animals' well-being.

For the assessment of welfare and well-being of animals in factory farming and food production, animals in private husbandry, animal-assisted therapy, zoos, wildlife or animal experiments, we need adequate techniques for an objective measurement of animal welfare (*Barnett & Hemsworth, 1990*) and associated physiological states as for example EEG, because, judging about animal welfare is most often based on or at least influenced by human assumptions and humanization (*World Organization for Animal Health (OIE), 2010*). It has been shown that observers who are "used" to expressions of horses that are associated with pain or stereotypic behaviors tend to underestimate these signs regarding horses' welfare (*Lesimple & Hausberger, 2014*). In a very recent review article, the authors explicitly mention EEG as potential tool to assess cognition and welfare, which are strongly associated (*Hausberger et al., 2019*). Especially in horses, which are most often kept as working animals, husbandry systems as well as education and training of the animals impact the welfare state. Nevertheless, behavioral data do not always represent the internal state of the animal, as has been shown in a comparative behavioral study with Chilean working horses and Rodeo horses (*Rosselot et al., 2019*). The need for an objective assessment of animal welfare is clearly apparent. Several authors proposed EEG as promising tool, but it is necessary to show that EEG is significantly correlated with behavioral data and to determine how to best analyze relatively complex skin derived EEG data to assess subtle changes.

We addressed these questions by combining a well-established technique for pain and stress assessment, the horse grimace scale (HGS) (*Dalla Costa et al., 2014*, *2018*), with telemetric EEG recordings as a promising novel tool for the measurement of objective data related to animal welfare and signs of stress (*Senko et al., 2017*; *Hohlbaum et al., 2017*; *Häger et al., 2017*). Facial expression scores are well-established to measure pain in human infants (*Grunau & Craig, 1987*) and for some nonhuman species with clear facial expression, such as the horse grimace scale (HGS, *Dalla Costa et al., 2014*), the mouse grimace scale (MGS, *Langford et al., 2010*) and the rat grimace scale (RGS, *Sotocinal et al., 2011*). *Dalla Costa et al. (2014)* validated the facial expression score by using a statistical approach to identify a classifier that can estimate the pain status of the animal based on Facial Action Units. There exists no doubt that animals are able to experience pain, fear, stress and other moods and show these through facial expressions. The electroencephalogram (EEG), first described by *Berger (1929)* is a method to measure tiny summed electrical potentials on the scalp surface that arise from pyramidal cells of the cortex. Therefore, noninvasive EEG measurements always represent network activity of cortical neurons rather than single cell activity. Cortical networks are highly dynamic and they are broadly orchestrated, which leads to electrical oscillations that can be measured on the surface of the scalp (*Kida, Tanaka & Kakigi, 2016*). These oscillations are

in the range of very slow waves below 0.1 Hz, as they occur for example in preterm babies (*Vanhatalo et al., 2002*). Faster oscillations are categorized as theta, alpha, beta and gamma bands. Some authors additionally define waves above 90 Hz as a "high gamma EEG-band" (*Cavelli et al., 2017*). A shift from low frequency EEG activity towards high frequency, low amplitude activity has been described during the castration procedure of calves (*Coetzee, 2013*). A shift of EEG band activity and lateralization have also been observed during attention-related processes in horses (*Rochais et al., 2018*). In conclusion, EEG data can represent behavioral or internal states of the animal.

More than 20 years ago, it was hypothesized that new tools of EEG analysis will lead to objective measurements for the assessment of animal welfare (*Klemm, 1992*). One method to find qualitative changes of network activity is phase amplitude coupling (PAC), also known as cross frequency coupling (*Tort et al., 2010*). PAC is based on amplitude modulations between EEG bands if they occur in a certain phase relation to each other (*Tort et al., 2010*). This phenomenon does, for example, occur during certain vigilance states (*Scheffzük et al., 2011*) or it can be modified by the application of drugs (*Scheffzük et al., 2013*). Therefore, we used, in a proof of concept experiment, the HGS as well as the PAC as a qualitative network change index and compared the intensity of amplitude modulation under resting condition and during stress condition (i.e., anticipating a medical treatment of horses in the veterinary clinic). We hypothesized that the HGS as well as the PAC are significantly different in the horses in accordance to the different conditions and that there is an association between HGS and PAC, which means the EEG could be an objective tool for the assessment of animal welfare.

## METHODS

### Ethical approval

All procedures were approved by the local ethics committee (L0294113; LAGeSo, Berlin, Germany), and followed the European and the German national regulations (Animal Welfare act, 2010/63/EU). All animal procedures were performed in accordance with the animal care committee's regulations (Freie Universität Berlin).

## MATERIALS

We used a Panasonic Digital camera Lumix DMC-FZ200, disposable adhesive surface silver/silverchloride electrodes (Spes Medica S.r.l., 111, Genova, Italy), abrasive cream (Abralyt HiCl; Easycap GmbH, Herrsching, Germany) and an EEG telemetry unit (with an AC coupled amplifier, sampling rate 500 Hz) (*Lapray et al., 2009*).

### Subjects

The three adult horses belonged to the demonstration stock for veterinary students at the Freie Universität Berlin. The horses were adults, of different sex and diverse breedings (one trotter (a race horse), two warmblood (most often used for show jumping or dressage) horses). The mare had been used in three veterinary demonstration treatments, the trotter (gelding) in two and the warmblood gelding in one treatment. The horses were not used as working or sports horses at the university but they had unknown origin. All horses are

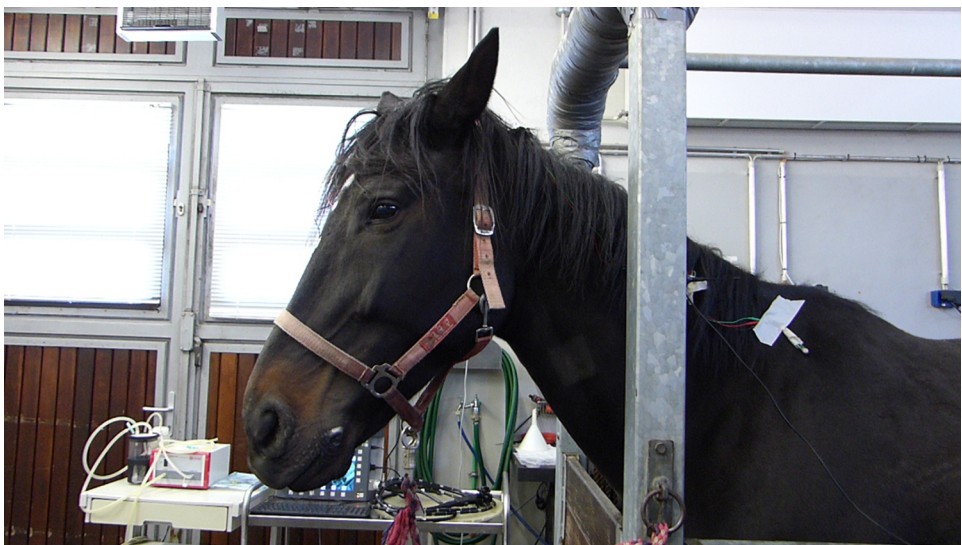

**Figure 1 Horse in examination stand.** The telemetry unit is fixed at the shoulder with a piece of tape. As an example, typical facial expressions in this picture are: Ears oriented backwards (score 2), closed eyes (score 0), tension above eyes (score 1), tension in the region around jaw muscles (score 1), tension around muzzle and prominent chin (score 1), tension around nozzles, flattened profile (score 2). The sum of all facial expression scores is 7 for this image.                     

healthy, used to handling and kept in stalls during the night and a paddock or pasture during the day. The experiments took place in their familiar environment at the horse clinic.

## Procedure

During six different days, three adult horses were recorded ($n = 6$ different veterinary interventions). All of our measurements took place during routine teaching lessons for the students. We measured two vigilance states of each horse for each experimental day (some horses have been recorded at two different veterinary treatments) with the HGS and with EEG. Both measurements took place simultaneously for about 30 min duration. The recordings were always performed by the same persons. One person did the video recordings, another person did the EEG recordings and a third person remained near to the horse to relocate the antenna, if necessary. The two experimenters, responsible for data acquisition (video and EEG), always started their new video and EEG files at the same time (each file approximately 10 min). First, the animals were recorded in a familiar situation in the stable to assess the relaxed state as a reference, called "resting condition". The second measurement took place in the examination stand in anticipation of stressful situation that is, a veterinary treatment, called "stress condition" (Fig. 1).

## Horse grimace scale

The video recordings were used for analyzing the HGS in accordance to *Dalla Costa et al. (2014)*. One HGS value was calculated for each video file. Regarding the correlation with EEG coupling coefficients, corresponding files were selected, but the video files were of longer duration than the EEG files. To calculate the total pain score with the HGS, six facial

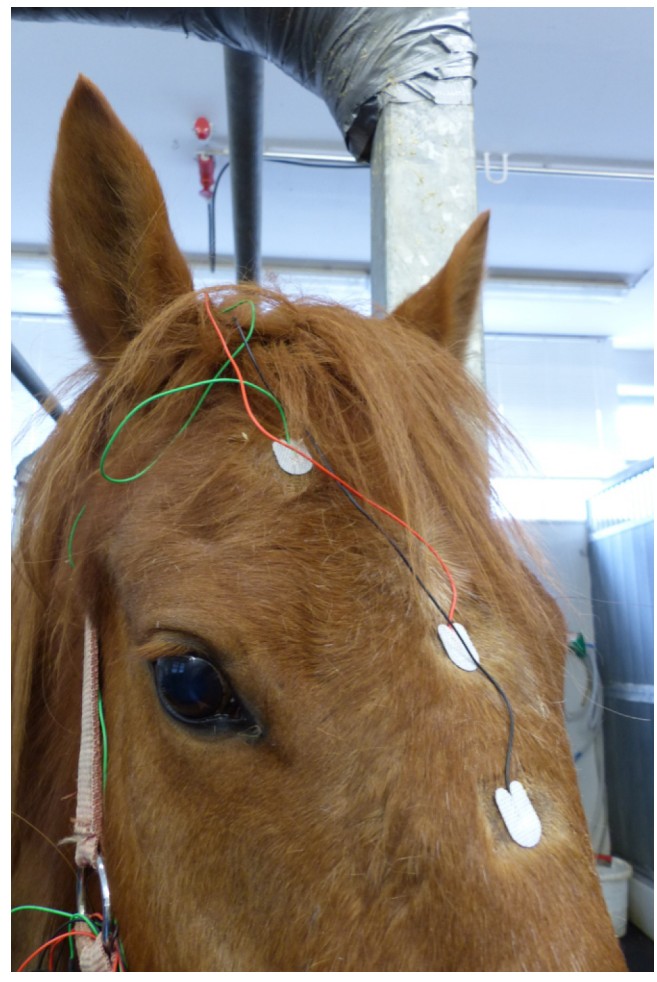

**Figure 2  Electrode positions.** The ground and reference electrode were placed above the nose (2, 3), the recording electrode was placed in a parietal position between the right eye and ear (1).

coding units were used (Ears stiffly backwards, orbital tightening, tension above eye area, prominent strained chewing muscles, mouth strained and pronounced chin, strained nostrils and flattening of the profile). For each coding unit a score of 0, 1 or 2 was given. This results in a maximal total pain score of 12 points. The HGS has a relatively high inter observer reliability. The Interclass Correlation Coefficient (ICC) has a value of 0.92 (*Dalla Costa et al., 2014*).

## EEG recordings

Disposable adhesive surface silver/silverchloride electrodes (Spes Medica S.r.l., 111 Genova, Italy) were placed on the nose (between the ears) as ground and reference (Fig. 2 (2, 3)) and between the eye and ear to record from the right somatosensory cortex region (parietal position, Fig. 2 (1)). Before fixating the electrode, the location was shaved and the skin was cleaned with an abrasive cream (Abralyt HiCl; Easycap GmbH, Herrsching, Germany) in order to remove dead skin cells and to achieve a lower impedance. The data

was recorded and sent by a telemetry unit (with an AC coupled amplifier, sampling rate 500 Hz) (*Lapray et al., 2009*). Only phases without artifacts were taken into account for analysis. Selected sequences lasted on an average 90 s in case of the resting condition and 50 s in case of the stress condition. For each veterinary intervention and each condition (rest or stress), only one sequence was used for analysis. Two horses have been treated two and three times, at different days (at least 1 week between the first and the second measurement).

## EEG analysis

EEG segments without artifacts (muscle, extended line noise) were selected.

We analyzed the data with Matlab (2016, MathWorks) and with Brainstorm (*Tadel et al., 2011*). EEG raw data were filtered with digital butterworth filters with a custom written Matlab script. The filter was designed with the function butter ($n$ = 3rd order). We calculated the normalized cutoff frequency (Wn) for EEG bands delta (0–4 Hz), theta (4–8 Hz), alpha (8–13 Hz), beta (13–30 Hz), low gamma (30–80 Hz) and high gamma (80–120 Hz). Wn is a number between 0 and 1, where 1 corresponds to the Nyquist frequency which is half the sampling rate (here: 500 Hz).

The numerator and denominator values (IIR filter), achieved with the function butter, were used with the Matlab function filtfilt to filter the EEG data. For the delta EEG band, a lowpass filter was used. We extracted all other EEG frequency bands with a bandpass filter design.

PAC and Phase Locking Value were analyzed with brainstorm software (*Tadel et al., 2011*).

To obtain Canolty maps (*Canolty et al., 2006*), the following procedure was computed (http://neuroimage.usc.edu/brainstorm/Tutorials/Resting): the EEG was filtered at the low frequency of interest, using a narrow band pass filter. The amplitude troughs of the desired low frequency were detected in the signal. A time window was defined around the detected troughs in order to compute a time frequency decomposition using a set of narrow band-pass filters.

## Analysis

To extract statistically significant changes of amplitude modulation, we compared the modulation indices calculated with the open source software brainstorm (*Tadel et al., 2011*) between the resting condition and the stress condition of the horses. With the brainstorm function "Canolty", the data are screened for amplitude modulations at higher frequency bands in relation to a certain slower phase frequency (here, we show the results for 8 Hz, Fig. 3). First, we pooled all cycles of the phase frequency (8 Hz) for each animal during a time window of 1 s, corresponding to eight phase cycles, and used the sum for each animal. The 1 s time window is independent of the duration of raw data; it means that eight phase frequency waves are represented in the resulting Canolty-map computation by the program brainstorm. Furthermore, we were interested in the information, at which point in the phase frequency cycle a possible amplitude modulation takes place (e.g., at the trough or the up-stroke of the 8 Hz phase frequency wave). We had
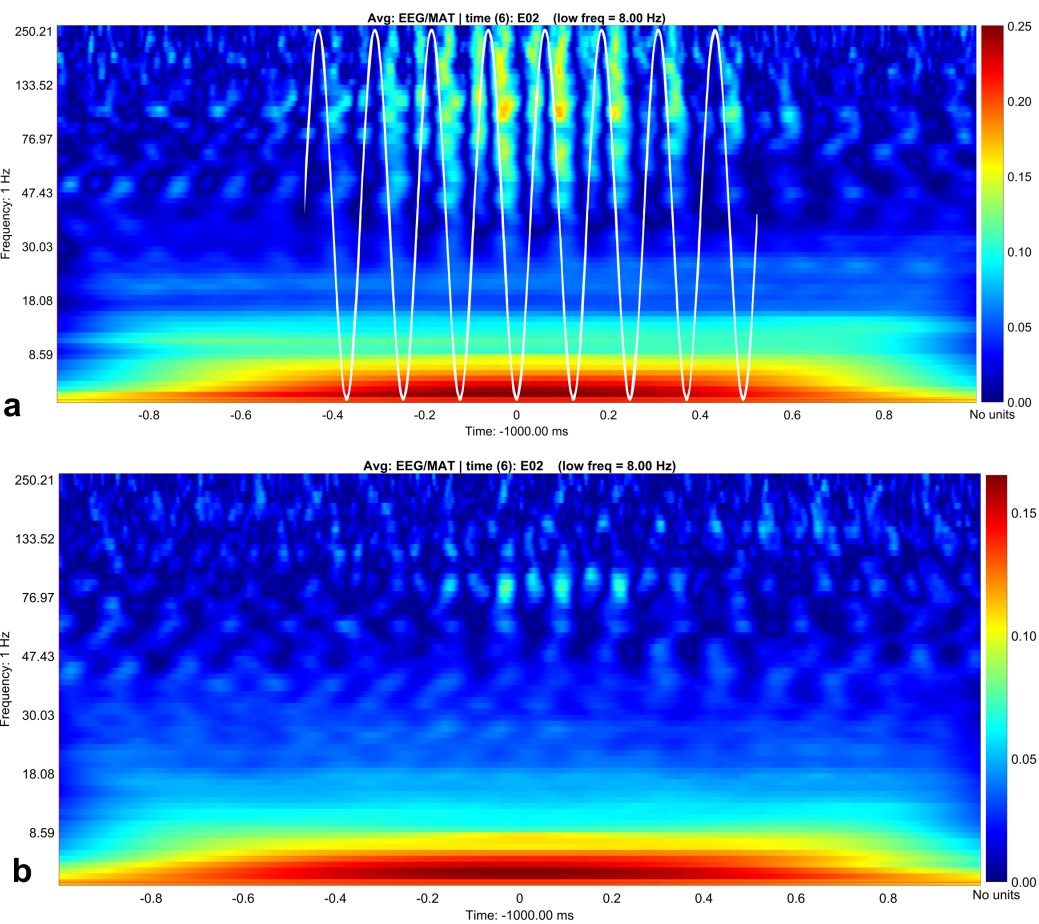

**Figure 3 Canolty maps for rest EEG and stress EEG.** Phase related amplitude modulation is shown as heat map. The low frequency for both behavioral conditions is 8 Hz (indicated as white sine wave in subplot A). Maximal amplitude modulation is visible during the up- and down stroke of the low frequency in the gamma and high gamma band with a peak at 130 Hz during rest (subplot A). In the stress condition (subplot B) the coupling pattern is quantitatively weaker, additionally the phase relation is changed. Amplitude modulation is maximal during the down stroke, the upstroke is only weakly associated with amplitude modulation in contrast to the resting condition (subplot A). Maximal modulation index values are again visible around 130° in the high gamma range.

a resolution of 125 modulation index values (given by the program brainstorm with the function "Canolty") within a single cycle of the phase frequency, which corresponds to 360° of the 8 Hz phase frequency wave. Ten millisecond in the Canolty-maps correspond to 2.9° of the phase frequency. We calculated the differences of modulation indices for every 2.9° of the phase frequency between the resting and the stress condition. The statistically significant results are shown as polar plot (number of statistically significant differences in relation to 360° of the phase frequency, Fig. 4). Furthermore, we calculated the coefficient of correlation between the modulation index of the EEG and the results of the HGS for both behavioral states (function "corr", type "Spearman"; MatLab version 2016b; MathWorks, Natick, MA, USA). For each kind of analysis, we used timely corresponding files of EEG and video. The sequences taken into account are longer

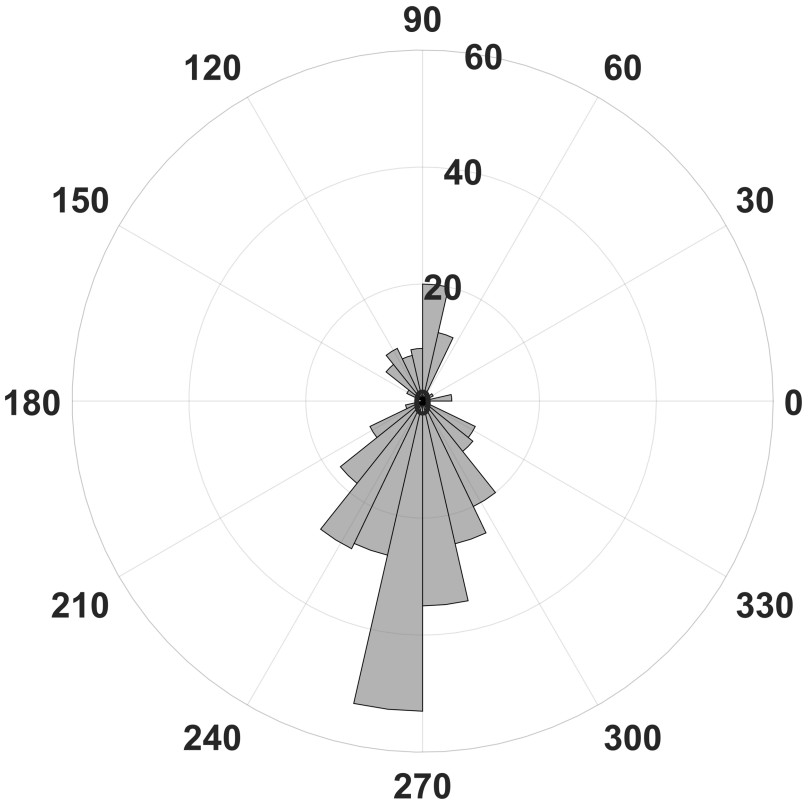

**Figure 4 Number of statistically significant differences of phase related amplitude modulation between rest and stress.** Most differences of the EEG derived modulation index between behavioral states can be found for the upstroke of the phase frequency around 90° and the down stroke around 270°.

for the video files than for the EEG, because, in the case of the EEG, we chose sequences without artifacts (most often artifacts from mobile phones in near vicinity).

We tested data for distribution with the Lilliefors test (Matlab, 2016b). Tests were performed with the non-parametric Kruskal Wallis test and a subsequent multiple comparison test in order to achieve exact statistical relations between groups. All tests are implemented in the Matlab statistics toolbox (Matlab, 2016b, MathWorks).

## RESULTS

We graded the horses' comfort behavior according to the HGS (*Dalla Costa et al., 2014*). EEG data and behavioral data were analyzed independently by two different persons to avoid a statistical bias. Both methods reflect changes of behavioral state. The HGS has a relatively high inter observer reliability. The ICC has a value of 0.92 (*Dalla Costa et al., 2014*).

### Horse grimace scale

We were able to identify statistically significant changes of the facial expressions between the resting condition and the stressful condition in horses ($p = 0.006$, confidence interval [0.489 3.258]). The mean facial expression score for the resting condition is

4.12 (with a standard deviation of 0.57 and an upper and lower confidence interval of 5.42 and 2.83). The mean facial expression score for the condition stress is 6 (with a standard deviation of 0.61 and an upper and lower confidence interval of 7.34 and 4.66). Furthermore we calculated the effect size with an $F$-test with $n = 3$, $F$ value = 6.383 and $p = 0.002$, which additionally shows that the facial expression is highly dependent on the behavioral condition (the third behavioral condition is stress under sedation, which was not used for EEG analysis).

## EEG

We were not able to detect statistically significant changes of horses' EEG band power between the rest condition and the stress condition in anticipation of a veterinary treatment. Nevertheless, there is a tendency towards slightly elevated EEG band power during the stress condition in comparison to the rest condition.

In contrast, we were able to identify major changes of PAC between the rest, and the stress condition (Fig. 3). A progressive decay of PAC between 8 Hz low frequency and the gamma and high gamma band takes place from rest towards stress. Besides the mere decay of coupling density, a qualitative change regarding phase relation between the high and low frequency is clearly visible. For the rest condition, the coupling patterns extend from gamma to high gamma (up to 250 Hz) with maximal coupling coefficients during the up stroke and the down stroke of the 8 Hz low frequency (indicated as white sine wave in Fig. 3A). For the condition stress (Fig. 3B), the PAC pattern is still visible in the gamma range but overall coupling strength is lower in comparison to the resting state. Statistically significant differences of phase related amplitude modulation between the rest and the stress condition can be found during the upstroke (around 90° of the phase frequency) and during the down stroke (around 260°) (Fig. 4), confirming the optical impression from the Canolty maps (Table S1). Furthermore, only the down stroke of the 8 Hz phase frequency reveals strong coupling coefficients with gamma and high gamma EEG bands. The coefficient of correlation (Rho) between the EEG derived data (Coupling coefficients of the Canolty map) and the HGS is $-0.86$ ($p = 0.00031$) for the down stroke of the phase frequency (here at 290°) and 0.71 ($p = 0.01$) for the up stroke of the phase frequency. The quality of raw data is good, the presence of nested activity (faster frequencies on top of slow frequencies) was confirmed with a wavelet analysis (Figs. 5A and 5B).

## DISCUSSION

Why assess welfare in horses? The horse is both companion as well as working animal (*Hausberger et al., 2019*). Facial expressions associated with pain and stress are well described (*Dalla Costa et al., 2014*) and relatively easy to assess for a trained observer. Horses raise several issues of animal welfare status, as, for example, regarding training methods, sports, as working animals and even as companions (boredom) (*Hausberger et al., 2019*). To judge about the validity and utility of EEG measurements in the context of animal welfare, horses are perfectly suited because they are used to handling procedures (which reduces the impact of the EEG application procedure itself on the subjects'

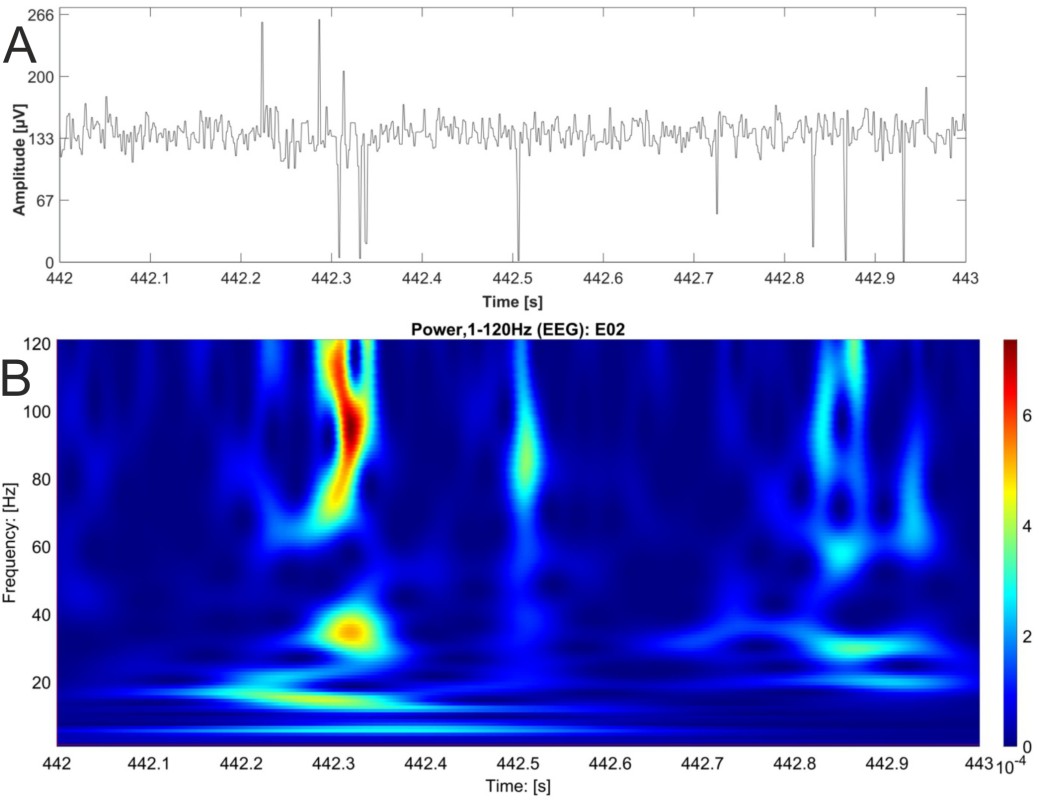

**Figure 5 EEG raw data trace with nested activity of multiple bands.** One second of original EEG recording is shown as raw data trace (A) and wavelet analysis (B). High power is coded as warm color (red). The wavelet analysis reveals nested activity in the alpha, beta and gamma band of the EEG.

behavior) and their facial expressions are well defined by the HGS, which we used as comparative value.

As has been proposed earlier (*Klemm, 1992*), we were able to extract differences in EEG network activity during stress and rest in horses, mental states that are obviously characterized by significant changes in HGS. In order to validate the behavioral states "stress" and "rest" in horses, we used the HGS (*Dalla Costa et al., 2014*, *2018*). PAC seems to be an extremely robust tool to extract activity patterns in the brain, associated with distinct behavioral states to assess animal welfare. One reason for the robustness might be that PAC is extracting information that is inherently linked to intrinsic neural network phenomena. The measurement does not rely on absolute values, which may be influenced by technical or intra-individual issues, but on a relational modulation index. Furthermore, phase related amplitude modulations seem to be an ubiquitous phenomenon for neural networks, as they are also involved in the generation of peristaltic movements of the intestinal tract (*Huizinga et al., 2015*). In fact, PAC in the intestinal tract is also associated with the regulation of peristaltic movements according to incoming stimuli (e.g., food intake). Further studies with a higher number of animals and more electrode positions may be beneficial to confirm our results and to gain further insights regarding anatomical correlations of certain behavioral representations in the EEG. Here, we used one electrode

in a parietal position. This position is near to the somatosensory cortex, a well characterized cortical region which is responsible for the representation of one's own body. Other electrode positions may give different results, as, for example, frontal electrode locations and should be addressed in subsequent studies.

Our results indicate a change of network activity during stress in the region of the somatosensory cortex. The progressive decay of coupling strength between the rest and the stress condition, with a maximum at half way of the down stroke (260°, Fig. 4), rather than an abrupt change, may be due to the general wave characteristics of envelope-signals, which are recorded from bigger neural networks. We can only speculate that the underlying entrainment of smaller sub-networks may work cascaded to form the progressive decay. A combination with noninvasive imaging techniques, as, for example, MRI or fMRI would be interesting to gain further insights regarding the location and regional dynamics of network entrainment. However, the technique is extremely challenging for behaving horses.

It can be excluded that physical pain is the main driver for the change of neural network activity because we took all stress recordings in the anticipation state, before any kind of physical pain or sedation appeared. The representation may look different in other brain regions. The mere touching or handling of the horse or the presence of people can be excluded as influencing factors, because these factors appeared in both conditions. Since the brain is more or less a black box in the case of noninvasive EEG measurements, it is very difficult or maybe impossible to reason about the origin of the coupling pattern on the cellular level with our method. The strong kind of stress reaction, which is probably associated with a medical treatment, may be relatively conserved among individuals. This may be different for minor forms of stress as, for example, during training, handling or social interaction. In these cases, we would expect a high intra-individual difference. To understand the change of PAC in relation to behavior in a better way, much more data are needed with different studies, different animals and different behavioral contexts.

Facial expression can hardly be used to recognize welfare in species like birds, reptiles etc. with poor facial expression. If it is possible to establish reliable EEG patterns associated with stress for different species in future studies, a translational approach for species with poor facial expression might be possible. It must be kept in mind that the activity of the brain is highly dynamic, also known as neural plasticity. The context can change the activity of the brain. Every measurement of the EEG can also potentially influence the content of the EEG—some kind of uncertainty relation. Telemetric EEG recordings are extremely useful in this context, because they do not restrict the freedom of movement of the animals. In this context, it must also be discussed whether the location (home stall during the rest condition versus treatment location during the stress condition) had an impact on the EEG. This possibility cannot be fully excluded. But, the home stall and the examination stand are located in the same building; furthermore, both locations are familiar to the horses. Of course, the examination stand might generally be associated with veterinary interventions for these horses because they are routinely used to train veterinary students for several diagnostic techniques. Therefore, the change of PAC patterns cannot be triggered by novelty (here: a novel environment) but the stress factor may arise

indirectly from the location, which is associated with stress rather than directly due to the veterinary treatment itself.

Since we did not use "experimental" horses exclusively for our study but demonstration horses of the university that were used for teaching purposes in a hands-on training for veterinary students, we were not able to select the horses in an optimal way for the purpose of the study. The number of training sessions was limited to six veterinary interventions at six different days with three different horses. It would be important to repeat this study with more horses and more observations.

Another important factor is the miniaturization of EEG amplifiers as well as comfortable electrodes that are as small as possible. For our future studies with a higher number of electrodes, we developed light curing polymer electrodes (*De Camp, Kalinka & Bergeler, 2018*). Furthermore, in a current project, the amplifier with eight Channels will be miniaturized to $9 \times 12$ mm. This EEG system will be potentially useful to assess EEG measurements in small species like birds or rodents. Another system is considered as a stackable (bus-) system, to include other bio-signals as for example ECG or breathing. Welfare monitoring may gain robustness by integrating multi-modal data.

## CONCLUSIONS

We hypothesized that the HGS as well as the modulation index PAC derived from EEG data are significantly different in the horses in accordance to different behavioral conditions and second, that there is an association between HGS and PAC, which means the EEG could be an objective tool for the assessment of animal welfare.

In conclusion, we were able to find an association between the scientifically validated HGS and an EEG pattern, associated with two distinct behavioral states, namely stress and rest in horses. PAC might be a robust tool for the objective assessment of animal welfare and well-being of animals. Furthermore, it may be useful to judge about brain states as well as comfort of neonates or disabled persons, who are unable to communicate actively.

Our pilot study gives preliminary evidence that EEG measurements can be an objective tool for the assessment of animal welfare and stress.

## ACKNOWLEDGEMENTS

We would like to thank Prof. Heidrun Gehlen, PD Dr. Dr. Ann Kristin Barton and Sabita Stöckle.

### Funding

Support was provided by the German Research Foundation (DFG) and the Open Access Publication Fund of Humboldt-Universität zu Berlin. The funders had no role in study design, data collection and analysis, decision to publish, or preparation of the manuscript.

## Grant Disclosures

The following grant information was disclosed by the authors:
German Research Foundation (DFG).
Humboldt-Universität zu Berlin.

## Competing Interests

The authors declare that they have no competing interests.

## Author Contributions

- Nora V. de Camp conceived and designed the experiments, performed the experiments, analyzed the data, prepared figures and/or tables, authored or reviewed drafts of the paper, and approved the final draft.
- Mechthild Ladwig-Wiegard conceived and designed the experiments, authored or reviewed drafts of the paper, and approved the final draft.
- Carola I.E. Geitner conceived and designed the experiments, performed the experiments, analyzed the data, prepared figures and/or tables, and approved the final draft.
- Jürgen Bergeler conceived and designed the experiments, performed the experiments, authored or reviewed drafts of the paper, and approved the final draft.
- Christa Thöne-Reineke conceived and designed the experiments, authored or reviewed drafts of the paper, and approved the final draft.

## Animal Ethics

The following information was supplied relating to ethical approvals (i.e., approving body and any reference numbers):

All procedures were approved by the local ethics committee (L0294113, Berlin, LAGeSo), and followed the European and the German national regulations (Animal Welfare act, 2010/63/EU). All animal procedures were performed in accordance with the animal care committee's regulations (Freie Universität Berlin).

## Data Availability

Matlab raw data files for the coupling coefficients of Canolty maps, computed via brainstorm, are available in the Supplemental Files.

## Supplemental Information

Supplemental information for this article can be found online at http://dx.doi.org/10.7717/peerj.8629#supplemental-information.

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
