# Peer review of "EEG based assessment of stress in horses: a pilot study"

_PeerJ, doi:10.7717/peerj.8629_

## Round 0.1 · original submission · Major Revisions

First let me apologize for the extremely longer than usual time to render a decision on your manuscript. As you can appreciate, very few reviewers have relevant expertise in both EEG assessments and equids. A second reviewer did agree but never came through with a review. Thus, I am basing the decision on one review and my own detailed reading of your paper. We converge on the impression that your data, as a preliminary report, is of value and can contribute to the literature, although it is clearly akin to pilot work. Please be sure to present the paper as such in a revision. I should note that I am not an expert in EEG; however, I do have interests in welfare assessments. Both the reviewer and I very much appreciate a novel technique for assessing welfare in animals, which is welcome given that no existing method is completely satisfactory. The reviewer points to the strengths of your rationale. I believe you could state it more explicitly in the revision.

Please break up the long paragraph on the second page of the MS. You could start a new paragraph on line 57. The introduction needs some reorganization. You should begin with the broad question – to assess welfare states in horses. The beginning of the discussion belongs in the Introduction. Describe existing methods and their limitations. Introduce the face pain grimace method as viable, and lastly, introduce EEG as a promising new direction. Define and describe it before discussing its particular application to the study of horse welfare. There should be at least some discussion as to why we might be concerned with horses’ well-being and in which contexts. You should review some of the prior work on well-being in horses.

You need sub-headings under Methods for Subjects, Materials and Procedure. You need to provide more detail on the history of the horses in your study. There is not nearly enough detail on the procedure. When were the videos recorded in relation to the EEG recordings? For how long were the horses recorded? Who did the recording and performed the tests? Was it always the same person?

I worry that one observation in each of two conditions with an N of 3 is very limited for drawing generalizations. Your paper would be greatly strengthened by repeating the observations in additional familiar and stressful situations. I assume the conditions were confounded with time and familiarity with the procedure. This has to be seriously addressed.
What is the possible range of scores on the facial grimace score? What is the reliability for raters?

The statistics are not described in enough detail.

The photos are great.

Minor comments:

Please change statistic to statistical on line 43 of p. 2.
Please check punctuation carefully especially for semi-colons and commas.
On line 57, pluralize horses, on pg. 2.
On line 141, place an ‘ after horses.
On line 142-143, I don’t think you mean statistical interference. I think you mean biases.
On line 178, change “lead” to “led.”

Reviewer 1 ·

Basic reporting

.

Experimental design

.

Validity of the findings

.

Additional comments

Review of “EEG based assessment of stress in horses: A Pilot Study” by de Camp et al. submitted to PeerJ is an interesting and refreshingly new take on non-invasive assessment of mental states of horses in anticipation of stressful condition.

In this study, the authors make a compelling case for need to establish species specific assessment criteria for their mental wellbeing across animals such as horses. They also argue in favor of developing novel techniques and adapting available technologies for this purpose. In this pilot study, they make the case by their innovative use of telemetric 3 lead EEG recording and attempt to correlate it with a well-established behavioral measure called, horse grimace scale (HGS). The EEG recordings and HGS assessments were made under conditions of “rest”, when the horses were in their stables or home environment, contrasted with “stress” condition, when they were taken for medical treatment. Their findings, though preliminary, show that there was significant change in HGS and cross-frequency coupling in EEG signal between the two conditions. As a proof of concept, while their results should be taken with caution, still provide promising trend that should be easily verifiable and may lead to significant insights into physiological read outs of the internal states of horses.

The experiments and data were collected following adequate protocols and care. Though they lack technical sophistication of standard physiological recordings, it is worth noting that these were done during free behaving condition and under minimal or no restraint to the animal, which could be quite challenging.

Here are a few questions and comments for the authors to help improve the manuscript:
1. Please provide rationale for choosing right somatosensory region for EEG recordings? Will the results be different if any other region on the head was selected for recordings?
2. Under EEG recording section, authors mentions “Selected sequences lasted on an average 90s in case of the resting condition and 50 s in case of the stress condition.. “ - How many such sequences occurred in a typical recording session? How many times the same individual was recorded under two conditions?
3. Any speculation with regard to the changes they saw in HGS and PAC, whether they were due to treatment stress or change in environment e.g. clinic vs home environment (change in local environment) OR treatment anticipation stress vs resting state.
4. Authors should discuss EEG changes more specifically that are related to their own data, for instance - a) their interpretation of the observed differential coupling strength between rest and stressful condition during up and down phases? b) intriguing but potentially interesting finding of a “progressive change in coupling strength” around ~8Hz and gamma band, rather than an abrupt change?
5. In future studies it will also be interesting to relate cross-frequency coupling changes in low and high gamma band using multielectrode EEG with change in heart rate of the animal in different conditions of stress.

Minor points:
Remove commas and replace with standard decimal notations when providing data and statistics – such as p=0,0006 should be changed to p=0.0006. This should be done throughout the manuscript.

Annotated reviews are not available for download in order to protect the identity of reviewers who chose to remain anonymous.

---

## Round 0.2 · Minor Revisions

Thank you for your revision. I have made my edits directly to your MS as there were a number of grammatical issues that needed correction. I have a number of other very minor requests for clarification. I can upload only a PDF file but I will ask that PeerJ make it accessible for me to send you the document in word file as well. With these minor changes, I think your study will make a modest but important contribution to the literature.

---

## Round 0.3 · accepted · Accept

Thank you for the revisions to the manuscript. I would just like to request that, during proofing, you insert a value for the stated high reliability of different raters on line 210.